# Artificial Intelligence in Transcatheter Aortic Valve Replacement: Its Current Role and Ongoing Challenges

**DOI:** 10.3390/diagnostics14030261

**Published:** 2024-01-25

**Authors:** Mina M. Benjamin, Mark G. Rabbat

**Affiliations:** 1Division of Cardiovascular Medicine, SSM—Saint Louis University Hospital, Saint Louis University, Saint Louis, MO 63104, USA; 2Department of Cardiovascular Medicine, Loyola University Medical Center, Maywood, IL 60153, USA; mrabbat@lumc.edu; 3Department of Cardiology, Edward Hines Jr. VA Hospital, Hines, IL 60141, USA

**Keywords:** artificial intelligence, TAVR, cardiology, risk prediction, procedure planning, monitoring

## Abstract

Transcatheter aortic valve replacement (TAVR) has emerged as a viable alternative to surgical aortic valve replacement, as accumulating clinical evidence has demonstrated its safety and efficacy. TAVR indications have expanded beyond high-risk or inoperable patients to include intermediate and low-risk patients with severe aortic stenosis. Artificial intelligence (AI) is revolutionizing the field of cardiology, aiding in the interpretation of medical imaging and developing risk models for at-risk individuals and those with cardiac disease. This article explores the growing role of AI in TAVR procedures and assesses its potential impact, with particular focus on its ability to improve patient selection, procedural planning, post-implantation monitoring and contribute to optimized patient outcomes. In addition, current challenges and future directions in AI implementation are highlighted.

## 1. Introduction

Transcatheter aortic valve replacement (TAVR) has emerged as a transformative approach, revolutionizing the management of severe aortic valve stenosis. The expanding indications for TAVR now include intermediate- and low-risk patients, reflecting its increasing acceptance and proven benefits across a wider spectrum of patients. Long-term follow-up studies, such as the PARTNER 2 [1] and SAPIEN 3 [2] trials, have confirmed the durability and sustained clinical benefits of TAVR, establishing it as the standard of care for aortic valve replacement in many clinical scenarios. According to national registries and large-scale studies, the number of TAVR procedures performed globally has shown a substantial increase, outpacing surgical aortic valve replacement [3].

The field of cardiology is witnessing significant advancements driven by the integration of artificial intelligence techniques in various domains. Artificial intelligence, powered by advanced computational capabilities and efficient data processing, has opened new avenues for improving care and reducing costs in medicine. Machine learning encompasses a broad range of techniques that enable computers to learn from data and improve their performance over time, without being explicitly programmed. Statistical methods commonly used in machine learning include regression, classification, clustering, hypothesis testing, cross-validation, and feature selection. Increased availability of large sets of data and developing computing power has led to a surge in supervised machine learning approaches with a wide range of potential applications, including the identification of potential therapeutic targets and drug development. Deep learning, a branch of machine learning, focuses on training deep neural networks, which are models inspired by the structure and function of the human brain. These deep neural networks consist of multiple layers of interconnected nodes, called neurons, that process and transform the data during learning. Within the realm of cardiology, artificial intelligence plays a significant role by assisting in the interpretation of medical imaging, such as echocardiograms and cardiac MRIs [4]. Additionally, artificial intelligence-powered wearable devices and remote monitoring systems can continuously track cardiac parameters, providing real-time insights and facilitating remote patient management for better overall cardiovascular health. Moreover, artificial intelligence-driven predictive models can analyze patient data and identify individuals at higher risk of cardiac events, enabling early intervention and better preventive care strategies. The application of artificial intelligence in TAVR procedures is being explored in order to improve patient selection and procedural planning and optimize patient outcomes and post-implantation valve monitoring.

In this article, we aim to highlight the growing role of artificial intelligence techniques and algorithms in TAVR procedures, and specifically focus on the following areas:Making a diagnosis of severe aortic stenosisPatient selectionProcedural planning and executionPredicting post-procedural morbidity and mortality

We also added a comprehensive limitations section, cautioning against the current optimistic hype surrounding the future role of artificial intelligence in clinical medicine, on the grounds this may lead to the premature adoption of unvalidated artificial intelligence algorithms in real-world settings, potentially compromising patient safety and quality of care. In considering the role of artificial intelligence in TAVR procedures, we also highlight current challenges and future directions.

## 2. Diagnosis of Severe Aortic Stenosis

Diagnosing severe aortic stenosis can be challenging because of its often asymptomatic nature in the early stages and overlap with the symptoms of other cardiovascular conditions. Traditionally, an increased gradient was used to diagnose severe aortic stenosis; however, fairly recently, an additional category of severe aortic stenosis has been identified in patients with low flow, defined as a stroke volume indexed to body surface area ≤35 mL/m^2^. These patients might not meet the cutoff gradient for severe aortic stenosis, but their calculated valve area will be in the severe range (i.e., <1 cm^2^). An accurate diagnosis is crucial as it enables timely intervention and appropriate management, preventing disease progression and improving patients’ quality of life and long-term prognosis. Although transthoracic echocardiography is the primary test for diagnosing and evaluating severe aortic stenosis, additional testing might be needed (including dobutamine stress echocardiography, and, in certain situations, aortic valve calcium scoring), as we detail below.

Artificial intelligence has shown that screening patients for severe aortic stenosis can even begin with electrocardiograms (EKGs), which are not typically used to diagnose aortic stenosis but are instead generally performed as a component of the initial evaluation. The main value of the EKG in this setting is to detect concomitant conditions such as atrial fibrillation and coronary disease, although similar repolarization abnormalities are caused by left ventricular hypertrophy and ischemia. Kwon et al. developed a deep learning-based algorithm by using over 39,000 EKGs which used the T-wave axis in leads V1–V4, the QT interval, and the patient’s age, to detect moderate or severe aortic stenosis with an area under the curve (AUC) of 0.861 [5].

The echocardiographic exam in patients with aortic stenosis includes the evaluation of valve anatomy and structure, valve hemodynamics, hemodynamic consequences (such as left ventricular hypertrophy or pulmonary hypertension), and concomitant aortic insufficiency; other cardiac valves are also evaluated during the echocardiogram. The echocardiographic exam is also important to determine eligibility for TAVR, provide guidance during the procedure, and assist follow-up after the procedure. A major issue in the echocardiography lab is the interobserver variability, which arises due to differences in experience, training, and subjective judgment between readers, leading to discrepancies in diagnostic results. Aortic stenosis is notoriously prone to interobserver variability due both to the previous factors, and also a multitude of parameters that contribute to the computation of the aortic valve area. Artificial intelligence has shown great potential in various applications within the field of echocardiography, offering a promising solution to address issues related to interobserver variability and improving efficiency in busy echocardiography laboratories. By assisting with echocardiography interpretation, artificial intelligence can also play a vital role in situations or locations where qualified individuals may not be readily accessible. Deep learning models have demonstrated efficacy in identifying left ventricular hypertrophy, which is associated with aortic stenosis, and also in accurately measuring left ventricular volumes and dysfunction [6]. Ventricular strain refers to the extent of deformation or stretching experienced by the myocardium during the cardiac cycle. It is a measure of the mechanical stress placed on the ventricular walls and has been shown to detect subtle myocardial damage that has not manifested as a decline in the ejection fraction and can also be used to evaluate adverse ventricular remodeling. Strain can be measured by echocardiography or cardiac magnetic resonance imaging. Cardiac magnetic resonance has proven to be a viable alternative to echocardiography when the quality of echocardiograms is inadequate. Cardiac magnetic resonance provides highly detailed multi-planar images, allowing for comprehensive visualization of the aortic valve and surrounding structures. Unlike echocardiography, cardiac magnetic resonance is not limited by acoustic windows, and provides clearer views, even in patients with challenging anatomy. Compared to CT, cardiac magnetic resonance sequences provide enhanced temporal resolution, albeit at the cost of a more extended scanning time [7,8].

The use of CT calcium scoring of the aortic valve can be useful in cases where echocardiographic results are equivocal or when low-flow, low-gradient severe aortic stenosis is present. The threshold used to identify true severe aortic stenosis is ≥1200 AU for women and ≥2000 AU for men. The severity of aortic valve calcification by CT is also predictive of mortality. Deep learning models that use CT scans to automatically segment and score aortic valve calcification have been developed [9], and some studies have aimed to derive aortic valve calcium scores from CTs obtained for other purposes, such as low-dose lung cancer screening or PET studies; these CTs however recorded suboptimal results, due to significant heterogeneity in their scanning protocols [10].

## 3. Patient Selection

One area where artificial intelligence is making significant contributions to TAVR procedures is in patient selection. In 2017, the European Society of Cardiology (ESC) guidelines expanded the TAVR patient collective to include intermediate risk patients and some low-risk patients who fulfill certain criteria that favor TAVR [11]. The 2020 American College of Cardiology/American Heart Association (ACC/AHA) guidelines marked a paradigm shift, in which the emphasis moved from a pure risk-related assessment to age- and durability-related considerations [12]. This change reflected the significant improvements in the transcatheter approach, reductions in intervention-related complications and studies of progressively lower risk patient groups. Hasimbegovic et al. used data from 532 patients who were enrolled in the VIenna CardioThOracic Aortic Valve RegistrY (VICTORY). They developed a machine learning-based approach for predicting the decision for TAVR versus surgical aortic valve replacement, which performed excellently (AUC 0.91 with 90% accuracy, 92% sensitivity and 90% specificity), demonstrating that machine learning can tap into the studying and understanding of complex clinical decision-making processes [13].

## 4. Pre-Procedural Planning

The pre-procedural process for TAVR entails screening for coronary artery disease, which now often uses coronary CT angiography. Artificial intelligence algorithms can model and predict coronary fractional flow reserve (FFR) values from CT angiography data, allowing for non-invasive assessment of the hemodynamic significance of intermediate lesions, leading to the safe deferral of unnecessary invasive coronary angiography and reducing complications [14,15,16]. CT, which is now recommended as an essential preliminary investigation for patients undergoing TAVR, provides ample anatomical detail for valve selection and sizing, and allows for the evaluation of potential complications, including the risk of annular rupture, which is crucial for ensuring patient safety. It also provides an assessment of peripheral vascular access, aiding the determination of the most suitable approach and reducing the likelihood of access-related complications. 

These scans are typically analyzed by a trained cardiologist or radiologist and are labor-intensive. Artificial intelligence algorithms have been developed for automating the critical steps from CTs, such as determining the aortic valve annular plane (Figure 1) and determining the access route (Figure 2). By leveraging machine learning techniques, these algorithms can measure the aortic annulus and aid device sizing with great precision, minimizing human error and optimizing patient-specific treatment strategies [17,18,19]. Santaló-Corcoy et al. developed TAVI-PREP, a fully automated deep learning-based method, for pre-TAVI planning. TAVI-PREP uses MeshDeformNet for 3D surface mesh generation and a 3D Residual U-Net for landmark detection. TAVI-PREP is designed to extract 22 different measurements from the aortic valvular complex. The mean absolute relative error was within 5% for most measurements, except for left and right coronary height [20]. During the TAVR procedure, an optimal view is essential when the X-ray tube C-arm is perpendicular to the aortic annulus plane, as this allows the operator to achieve appropriate delivery of the valve. Theriault Lauzier et al. developed a convoluted neural network to infer the location and orientation of the aortic valve annular plane. The proposed algorithm had accuracy on par with proposed automated methods for localization and approaches an expert-level accuracy. Here, note that the method is not specific to the aortic valve and may be generalizable to other anatomical features [21]. Samin et al. developed an automated method in which a 3-D model derived from cardiac CTs can be used to predict the best line of perpendicularity of the proposed TAVR valve, and thus provide the best C-arm angle for alignment of the new valve with the aortic valve annulus [22].

Cardiac magnetic resonance can also be used as an alternative to CT in pre-TAVR planning, and specifically for patients whose renal function prohibits iodinated contrast or who have experienced a severe allergic reaction to iodine. Recently developed cardiac magnetic resonance advances enable full 4D mapping of intravascular flows, and provide a crucial function that can be used to assess hemodynamics in patients with severe aortic stenosis [7,8].

Recently, computational models have been developed to potentially allow the virtual implantation of various device sizes at different implantation depths for a specific patient, with the goal of providing insights that are needed for physician decision-making and procedure planning. These simulations show blood flow profiles that can be used to assess hemodynamics after valve implantation, and also predict the mechanical stress imposed by the proposed valve on the tissue structures, such as contact pressure and wall shear stress on the aortic wall, along with principal stress on the aortic leaflets. Fluid structural interaction (FSI) computations are comprehensive and consider blood flow during the cardiac cycle, coupled with the structural mechanics of the valve [23]. However, these highly complex computations are considered to be impractical. Finite element analysis (FEA) and computational fluid dynamics (CFD) models, which are less complex, have also been investigated l. Finite element analysis focuses on determining the stent contact areas on walls, which is important in the assessment of anchoring. This simplified model type neglects blood flow and therefore does not reliably simulate the dynamics of the valve [24,25]. Computational fluid dynamics studies, on the other hand, consider blood flow dynamics, enabling assessment of valve function and the identification of potential paravalvular leakages [26]. Again, these techniques still require long computational times, which often renders them impractical for clinicians. There is great potential that deep learning approaches can expedite these models. Liang et al. proposed a model to directly estimate the stress distributions of the aorta for ascending aortic aneurysm patients, and developed a fully connected neural network that can compute stress distributions at a much faster rate than finite element analysis [27]. Similarly, fully connected neural network have been conceived and developed as a faster alternative to computational fluid dynamics and used to estimate the steady-state distributions of pressure and flow velocity inside the thoracic aorta, [28]. Balu et al. developed a deep learning-based model to learn about the deformation biomechanics of bioprosthetic aortic valves. A convoluted neural network was developed to predict the final deformed, closed shape of the heart valve from the input aorta geometry of the original undeformed heart valve [29]. Oldenburg et al. used a U-Net architecture to predict simplified 2D flow during peak systolic steady-state blood flow through mechanical aortic valves with varying opening angles in randomly generated aortic root geometries, achieving a validation error below 0.06. The neural network generates flow field prediction in real time, which is more than 2500 times faster than computational flow dynamics simulation [30]. These models are not currently ready for prime-time application, and more comprehensive models that are specific to TAVR need to be developed and applied on a larger scale before they can be incorporated into the daily workflow.

## 5. Predicting Mortality Risk

Predicting mortality before TAVR is of paramount importance as it allows clinicians to assess the potential risks and benefits of the procedure for individual patients, aiding in informed decision-making. Accurate mortality prediction helps optimize patient selection, enhance procedural planning, and improve patient outcomes by tailoring treatment strategies based on individual risk profiles. The most commonly used surgical risk assessment models for predicting outcomes in patients with severe aortic stenosis include the logistic EuroSCORE I, EuroSCORE II, and the Society of Thoracic Surgeons (STS) score, which all predict 30-day survival [31,32]. For TAVR, they are poor predictors of mortality that focus on procedural or 30-day mortality, and this also applies to the TAVR-specific TVT registry score [33]. The prediction of one-year mortality is even more challenging, and so TAVI2-SCORE and CoreValve models that are TAVR-specific were devised [34,35]. CAPRI risk scores, which based on the linear predictors of Cox models, including thoracic aortic calcification and comorbidities and demographic, atherosclerotic disease and cardiac function factors were also developed [36]. Artificial intelligence has shown promising results in predicting risk following TAVR by leveraging advanced algorithms to analyze a patients’ clinical data and provideimaging results and has, in so doing, outperformed traditional scoring systems. Evertz et al., showed the automated quantification of ventricular volumes and function was noninferior to human reader quantification that sought to predict cardiovascular mortality after TAVR. In addition, fully automated quantification also resulted in a time saving of ten minutes per patient [7]. Abdul Ghaffar et al. used a semi-supervised automated machine learning approach to classify patients into phenotypic groups, who were ordered by their estimated mortality. Their patient similarity network identified five patient phenogroups, and substantial variations in clinical comorbidities and in-hospital and 30-day outcomes. For 30-day cardiovascular mortality, the use of phenogroup data in conjunction with the STS score was found to improve the overall prediction of mortality, when compared against using the STS scores alone (AUC 0.96 vs. AUC 0.8, *p* = 0.02) [37]. Similarly, Gomes et al. studied 83 features of 451 consecutive patients who underwent TAVR and found machine learning methods were superior to STS and STS/ACC TAVR scores in predicting all-cause intrahospital mortality [38]. Another machine learning prediction model developed by Agasthi et al., which included 163 variables from 1055 TAVR patients, outperformed TAVI_2_-SCORE and CoreValve Score in predicting mortalityoneyear after TAVR (AUC 0.72 vs. 0.56 and 0.53) [39]. Hernandez-Suarez et al. conducted the second largest study to derive a risk prediction model frompatients who underwent TAVR, and found it was second to the STS/ACC TVT score. They developed a NIS-TAVR score from a cohort of over 10,000 TAVR patients with a 3.6% total mortality rate and developed a model that outperformed established risk calculators, with an AUC of 0.92 [40].

## 6. Predicting Specific Outcomes

Artificial intelligence models have also been developed to predict specific complications after the TAVR implantation. Length of stay following TAVR continues to improve, but significant gaps remain in meeting early discharge goals. Judson et al. used data from 9360 outpatient TAVR procedures, and developed a supervised random forest plot machine learning algorithm to identify variables involved in short (<36 h) and long (≥72 h) length of stay. The predictive power of machine learning models (AUC 0.82 and 0.85) was more robust than the standard multivariate model (AUC 0.65, and 0.65). Several novel predictors were identified in the algorithm, including procedural duration, need for post-procedure physical therapy, and procedure day of the week [41]. 30 days after TAVR, one in three readmissions was attributed to heart failure and associated with higher readmission mortality rates. Predicting heart failure admissions after TAVR holds would allow healthcare providers to proactively manage patients’ postoperative care, optimize treatment strategies, and allocate resources effectively. Khan et al. used data from 92,363 TAVR cases drawn from the National Readmission Database, and found a total of 3299 (3.6%) were readmitted within 30days with HF. A total of seven variables, based on predictive ability as well as clinical relevance, were selected and a performance evaluation that used the testing dataset achieved an AUC of 0.76 [42].

Evidence about the frequency and clinical significance of subclinical valve leaflet thrombosis of bioprosthetic valves following TAVR is emerging, and subclinical thrombosis has been found to be associated with significantly increased rates of transient ischemic attacks. Bailoor et al. conducted a computational proof-of-concept study and demonstrated the feasibility of using artificial intelligence algorithms for detecting reduced mobility in individual leaflets of prosthetic valves by using pressure measurements from microsensors embedded on the valve stent. By leveraging data-driven analysis and machine learning techniques, their artificial intelligence-based approach achieved an accuracy rate of over 90% in the prospective detection of leaflet dysfunction [43].

Cerebrovascular events are a potential complication of TAVR procedures. According to published registries, the overall incidence rate of stroke in high-risk patients after TAVR varied from 1.7% to 4.8%, compared with 0.5%–5.7% for surgical aortic valve replacement [44,45]. Okuno et al. developed a machine learning model by using data from 2279 patients to predict 30-day cerebrovascular events (AUC 0.79) [46], and Baig et al. developed a real-time autonomous intraoperative neuromonitoring tool that used transcranial doppler from patients undergoing TAVR [47].

Conduction abnormalities leading to the need for permanent pacemaker implantation is one of the most common TAVR postprocedural complications, with an incidence of up to 22% in some studies. TAVR results in an increased likelihood of native conduction system damage, owing to a combination of significantly greater patient comorbidity and the mechanism of TAVR deployment [48]. Truong et al. retrospectively studied 557 patients in sinus rhythm undergoing TAVR and developed machine learning models for assessing the likelihood of permanent pacemaker implantation. They incorporated data from pre- and post- TAVR EKGs, and also drew on clinical and echocardiographic data in their analysis. The random forest model performed better than the logistic regression model in predicting permanent pacemaker implantation risk (AUC: 0.81 vs. 0.69) [49].

The rate of bleeding complications is reported to be 3% to 11% within the first year of TAVR, with most episodes occurring early [50]. The median onset of late bleeding (>30 days) is 132 days in PARTNER registries. Bleeding is a strong independent risk factor for mortality for between 30 days and one year (adjusted HR 3.91, 95% CI 2.67–5.71) in the PARTNER randomized cohorts and continued access registries. The most frequent types of major late bleeds were gastrointestinal (41%), neurological (16%), and traumatic fall-related (8%) [1]. Navarese et al. incorporated more than 100 clinical variables from 5185 consecutive patients undergoing TAVR in the prospective multicenter RISPEVA (Registro Italiano GISE sull’Impianto di Valvola Aortica Percutanea) to develop a model whose performance was externally validated in 5043 TAVR patients by the prospective multicenter POL-TAVI (Polish Registry of Transcatheter Aortic Valve Implantation) database. The model uses six items to predict the 30-day risk of post-TAVR bleeding. External validation produced a 30-day AUC of 0.78 (95% CI: 0.72–0.82) [51]. Jia et al. developed a deep learning-based model named BLeNet with 56 features (covering baseline, procedural, and post-procedural characteristics) to predict post-procedural bleeding. The BLeNet model significantly outperformed the Cox-Proportional-Hazard and random survival forest models in discrimination and calibration. In the Kaplan-Meier analysis, the BLeNet model showed great performance in stratifying high- and low-bleeding risk patients (*p* < 0.0001) [52].

Left ventricular mass regression is an expected phenomena in patients who have undergone aortic valve replacement. This regression is reported to be tightly correlated with survival after the intervention, in years one and five. Asheghan et al. drew on CT data from 66 patients and statistical shape analysis techniques, and combined them with customized machine learning methods to extract latent information from segmented left ventricle shapes, which enabled them to predict left ventricular mass index regression a year after TAVR. The average accuracy of the predictions was validated against clinical measurements and used to calculate root mean square error and R2 score, which yielded values of 0.28 and 0.67, respectively, for the test data [53]. A summary of the machine models used for patient selection, procedure planning and risk assessment, are summarized in Table 1.

Cardiac magnetic resonance can also be used after TAVR to provide an accurate estimate of ventricular function, and to detect and quantify paravalvular leak [54]. Rapid ventricular pacing routinely performed during TAVR has been shown to lead to microcirculatory arrest. This propensity—at least theoretically—can lead to subendocardial ischemia and myocardial necrosis. Cardiac magnetic resonance is being explored to evaluate myocardial necrosis [55], strain [56] and wall motion [57] after TAVR procedures. All these cardiac magnetic resonance features now use deep learning algorithms and require minimal human processing [7].

**Table 1 diagnostics-14-00261-t001:** Examples of machine learning platforms and models developed for TAVR patient selection, procedure planning and risk prediction.

Reference	Study Cases	Comparison	Outcome	ML Model	Main Results
**Diagnosing severe AS**					
Kwon [5]	39,371 EKGs. 6453 for internal validation and 10 865 for external validation	none	More than moderate. AS confirmed by echocardiography.	DNN + CNN	AUC 0.884 (95% CI, 0.880–0.887) and 0.861 (95% CI, 0.858–0.863) for internal and external validation, respectively
Chang [9]	589 CTs (412 training, 40 validation and 137 testing)	Manually measured AV calcium volume and Agatston score	Accurate grading of AS severity	modified 3D U-net CNN	Agatston score accuracy of grading = 92.9%. AUC = 0.933 (95% CI 0.885–0.981), outperformed radiologist readers.
**Patient selection**					
Hasimbegovic [13]	532 patients from VICTORY Registry	Heart team decision	SAVR versus TAVR	ML-based 3-layer model	AUC 0.91 (90% accuracy, 92% sensitivity and 90% specificity)
**Pre-procedural planning**					
Santaló-Corcoy [20]	200 CTs (35 for training, and 17 for testing)	Manual CT measurement by an expert cardiologist using 3Mensio.	Correlation between manual and automated measurements	DL algorithms (MeshDeformNet) for landmark detection followed by segmentation	mean absolute relative error was within 5% for most measurements, except for coronary height (11.6% and 16.5%).
Theriault-Lauzier [21]	94 CTs of severe AS (K-fold cross-validation with K=)	Manually segmented AV annulus	Correlation between manual and automated measurements	recursive multiresolution CNN for localization of the AV annulus centroid	average out-of-plane localization error of 0.9 ± 0.8 mm for the evaluation dataset. The proposed algorithm is on par with automated methods for localization and approaches in providing an expert-level accuracy.
Samin [22]	60 CTs (24 retrospectively, 36 prospectively)	fluoroscopy	accurate prediction (<5° difference) of LP	Not detailed	Automated 3D analysis of CTs accurately predicted the LP aortic annulus and the corresponding C-arm position required in 8/8, 16/17, and 10/11 in patients with mild, moderate, or severe calcifications.
**Predicting mortality risk**					
Abdul Ghaffar [37]	354 TAVR cases divided into 2 cohorts	STS score	In-hospital and 30-day CV and all-cause mortality	TDA and a cloud-based supervised AutoML platform (OptiML)	The patient similarity network identified five patient phenogroups with substantial variations in clinical comorbidities and in-hospital and 30-day outcomes. Group 5 was associated with higher rates of 30-day CV mortality (OR 18, 95% CI 3–94), and 30-day all-cause mortality (OR 3, 95% CI 1.2–9).
Gomes [38]	Retrospective study of 451 TAVR cases	STS score	In-hospital and 30-day all-cause mortality; Secondary outcomes; Stroke, vascular complications; Paravalvular leak; and PPI	neural networks, support vector machines, and RF	performance of all MLmodels in predicting all-cause intrahospital mortality (AUC 0.94–0.97) was significantly higher than both the STS score (AUC 0.64), the STS/ACC TAVR score (AUC 0.65). Secondary outcomes could not be accurately predicted.
Agasthi [39]	1055 TAVR cases	TAVI2-SCORE and CoreValve score.	One-year mortality	GTB	AUC for GTB vs. TAVI2-SCORE and CoreValve Score was 0.72 (95% CI 0.68–0.78) vs. 0.56 (95%CI 0.51–0.62) and 0.53 (95% CI 0.47–0.59)
Hernandez-Suarez [40]	NIS database (2012–2015). (development: 7615, validation: 3268).	none	In-hospital mortality	Logistic regression, artificial NN, naive Bayes, and random forest	The prediction models showed good AUC performance AUC (>0.80). The best model was obtained by logistic regression (area under the curve: 0.92; 95% confidence interval: 0.89 to 0.95). Most obtained models plateaued after 10 variables were introduced.
**Predicting specific complications**					
Judson [41]	9360 cases from the BIOME dataset (2017–2021)	standard multivariate model	short length of stay (<36 h) and long length of stay (≥72 h)	RF	The predictive power, of both the short LOS (AUC 0.82) and long LOS (AUC 0.85) ML model, was more robust than the standard multivariate model (SLOS AUC 0.65, LLOS AUC 0.65).
Khan [42]	92,363 cases from National Readmission Database (2015–2018; 70% training, 20% validation, 10% testing)	none	30-day readmission for heart failure	“AutoScore” package, a ML-based automatic clinical score generator	AUC of TAVR-HF Score was 0.761 (95% CI 0.744–0.778)
Okuno [46]	2279 TAVR patients from Swiss TAVR registry (2/3 training, 1/3 test)	none	30-day Cerebrovascular events	ANN	The constructed model uses less than 107 clinical and imaging variables, and has AUC of 0.79 (0.65–0.93).
Truong [49]	557 cases-single center (75% training, 25% test)	logistic regression	Permanent pacemaker implantation (PPI)	RF	The RF model performed better than logistic regression model in predicting PPI risk (AUC: 0.81 vs. 0.69).
Navarese [51]	5185 cases from RISPEVA, validated in 5043 cases from the prospective POL-TAVI	PARIS and HAS-BLED scores	Major and minor bleeding within 30 days and 1 year	ML and univariate analyses were used for variable selection	The Optimism bootstrap-corrected AUC was 0.79 (95% CI: 0.75–0.83). Compared with PARIS and HAS-BLED, PREDICT-TAVR showed superior net benefit and improved predictive performance for all bleeding risk thresholds >2.5%
Jia [52]	668 cases-single center	Traditional Cox-PH and RF	Major or life-threatening bleeding	CNN	The BLeNet model outperformed the Cox-PH and RSF models [optimism-corrected c-index of BLeNet vs. Cox-PH vs. RSF: 0.81 (95% CI: 0.79–0.92) vs. 0.72 (95% CI: 0.63–0.77) vs. 0.70 (95% CI: 0.61–0.74)]
Lopes [58]	1478 cases-single center (70% training, 30% testing)	traditional logistic regression	One year mortality and improvement in dyspnea	SVM, RFC, MLP, and GTB	The RF classifier achieved the highest AUC (0.70) for predicting mortality. Logistic regression had the highest AUC (0.56) in predicting the improvement of dyspnea.

AS: aortic stenosis; AUC: area under the curve; AV: aortic valve; CI: confidence interval; Cox-PH: Cox proportional hazard; CV: cardiovascular; DL: deep learning; GTB: gradient tree boosting; LOS: length of stay; MLP: multi-layer perceptron; PPI: permanent pacemaker implantation; RFC: random forest classifier; SVM: support vector machine; TDA: topological data analysis.

## 7. Limitations, Challenges and Future Directions

While artificial intelligence holds immense potential for advancing TAVR procedures, challenges must be addressed to ensure its safe and responsible integration. Caution should be exercised regarding publication bias in artificial intelligence research, as studies with positive outcomes may be more likely to be published, leading to an overestimation of the technology’s validity and applicability [59]. As an example of the artificial intelligence models that have not outperformed logistic regression, the model developed by Lopes et al. did not have incremental value in predicting dyspnea in a population of 1478 patients who underwent TAVR, when compared to commonly applied likelihood ratio techniques [58]. Challenges also include data privacy, poorly selected/outdated data, selection bias, and unintentional continuance of historical biases/stereotypes in the data that can lead to inaccurate conclusions [59]. Though several clinical decision support models have been marketed, their safety, reproducibility, usability, validity, and reliability have caused concern [60,61,62]. Currently, there is an observed trade-off between the accuracy and interpretability of machine learning models. The numerous intertwined relationships captured by the layers of a deep neural network are only partially understood and the success of the models during implementation is not guaranteed. The interpretability of artificial intelligence models, particularly deep neural networks, remains an ongoing challenge, requiring further research and validation [63]. To mitigate publication bias, systematic reviews and meta-analyses that include both published and unpublished data are essential to provide a more comprehensive and balanced assessment of TAVR outcomes. The paucity of experienced reviewers in machine learning underscores the growing need for experts who can thoroughly vet and evaluate the technical aspects of machine learning models that have been proposed for publication. Their expertise is essential to ensure the quality, validity, and ethical implications of these models, and to foster responsible and impactful advancements in the field.

In the future, the models described in this manuscript will be validated and refined at more TAVR centers. Rigorous validation, regulatory scrutiny, and close collaboration between artificial intelligence developers and healthcare professionals are needed to ensure that artificial intelligence technologies are effectively integrated into clinical practice. Ultimately, we might anticipate deep learning models that integrate clinical data with imaging counterparts from echocardiography and computed tomography, work to conclude the feasibility of the procedure, identify the appropriate valve size, and assess the associated risk for different complication.

In conclusion, the advanced artificial intelligence algorithms described in this paper are promising tools that have, and will, further enhance the planning, execution and postoperative follow up of TAVR procedures. However, the authors also believe that nuanced clinical judgment by skilled physician teams will remain irreplaceable.

## Figures and Tables

**Figure 1 diagnostics-14-00261-f001:**
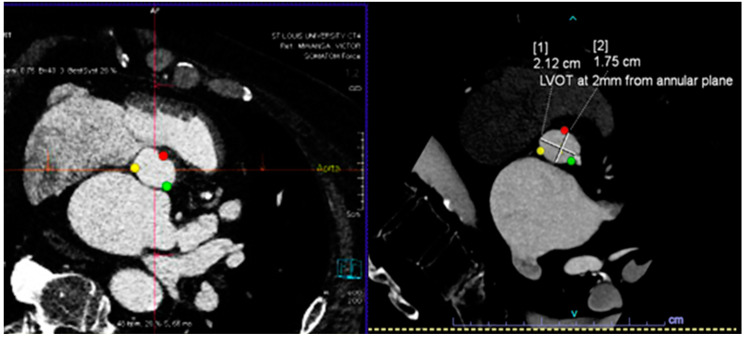
(**Top**) Automatic segmentation of the aortic valve annulus with three hinge points that are most evident on the left ventricular outflow tract side for each cusp (**top left**), and the left ventricular outflow tract plain that lies 2–4 mm below and is parallel to the aortic valve plain (**top right**). (**Bottom**) Automatic detection of the angle between the long axis of the aorta and the valve plain.

**Figure 2 diagnostics-14-00261-f002:**
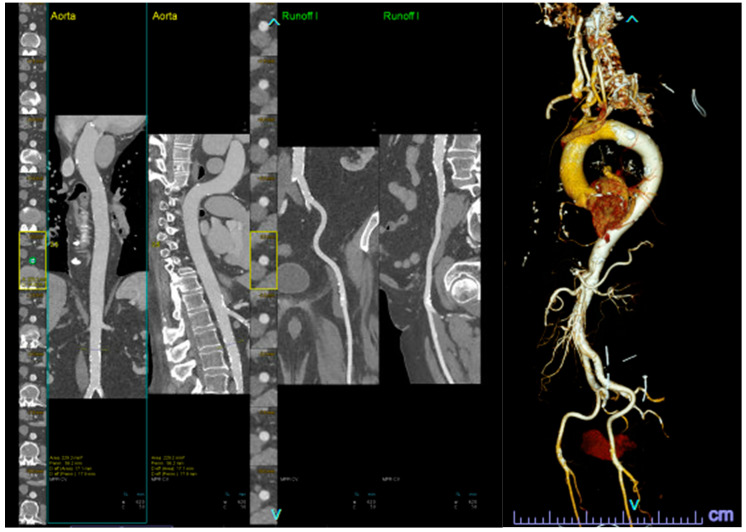
Automatic segmentation of the peripheral vasculature, both in 2-D (**left**) and 3-D (**right**), for TAVR access evaluation.

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
