# Peer review of "Artificial Intelligence in Transcatheter Aortic Valve Replacement: Its Current Role and Ongoing Challenges"

_diagnostics, 2024, doi:10.3390/diagnostics14030261_

Round 1
Reviewer 1 Report (Previous Reviewer 2)
Comments and Suggestions for Authors
This article explores the growing role of artificial intelligence (AI) in transcatheter aortic valve replacement (TAVR) surgery, from diagnosing severe aortic stenosis to selecting and preparing patients for the TAVR procedure. The authors also delve into the use of AI in surgical planning and predicting postoperative morbidity and mortality. They include a crucial warning about the current hype surrounding AI in clinical medicine, as it may lead to premature adoption of unvalidated AI algorithms in real-world settings, potentially jeopardizing patient safety and the quality of care.
This revised version has addressed the shortcomings of the previous draft. The authors have incorporated numerous quantitative comparisons from practical experiments, notably providing a detailed list associating the quantitative results in Table 1 with references. As a review paper, this version has reached the standard suitable for public dissemination.
Author Response
No issues to address. Thank you for reviewing.
Reviewer 2 Report (Previous Reviewer 3)
Comments and Suggestions for Authors
While the subject matter of this review paper is intriguing, the manuscript presents several drawbacks. It lacks a clear structure and fails to demonstrate a significant contribution toward advancing the current state of the art. Furthermore, numerous statements in the paper lack appropriate references to previous research findings, hindering the strength of the arguments presented.
The paper is riddled with numerous typos throughout its content.
There are duplicate Figure 1 with identical captions. Overall, the captions for the figures require improvement. The introduction lacks clarity regarding the contributions of this review paper. Utilizing bullet points in the introduction might aid in highlighting the paper's contributions.
In "A major issue in the echo- 95 cardiography lab is the interobserver variability which arises due to differences in experi- 96 ence, training, and subjective judgment between readers" I would advise clinicians and field experts instead of general readers.
I'd recommend enhancing the "Future Directions" section.
This manuscript lacks a conclusion section, a crucial part that allows authors to emphasize the contributions and motivations driving the necessity of the review paper.
Author Response
Thank you for reviewing our manuscript. Please find reviewer comments below and our responses below each comment in bold.
- While the subject matter of this review paper is intriguing, the manuscript presents several drawbacks. It lacks a clear structure and fails to demonstrate a significant contribution toward advancing the current state of the art. Furthermore, numerous statements in the paper lack appropriate references to previous research findings, hindering the strength of the arguments presented.
- The authors and the other two reviewers disagree that the manuscript lacks structure. A review paper is not an original research paper with new data and findings. The purpose of a review paper -rather- is outlining the current state of the art in a manner that is easy to assimilate for the readers while shedding light on current gaps and future directions.
- The paper is riddled with numerous typos throughout its content.
- The authors have proofread the manuscript one more time. The following corrections were made.
# “Expanding indications for TAVR now include intermediate- and low-risk patients” -> “The expanding indications for TAVR now include intermediate- and low-risk patients.”
# “Machine learning encompasses a broad range of techniques that enable computers to learn from data and improve their performance over time without being explicitly programmed” -> “Machine learning encompasses a broad range of techniques that enable computers to learn from data and improve their performance over time, without being explicitly programmed.”
# “Artificial intelligence has shown that screening patients for severe aortic stenosis can even begin with electrocardiograms EKGs" -> "Artificial intelligence has shown that screening patients for severe aortic stenosis can even begin with electrocardiograms (EKGs)"
# "The use of CT calcium scoring of the aortic valve can be useful in cases where the echocardiographic results are equivocal or in the context of low-flow, low-gradient aortic stenosis." -> "The use of CT calcium scoring of the aortic valve can be useful in cases where the echocardiographic results are equivocal or in the context of low-flow, low-gradient severe aortic stenosis."
# "To that end, TAVI2-SCORE and CoreValve models which are TAVR specific were devised." -> "To that end, TAVI2-SCORE and CoreValve models, which are TAVR specific, were devised."
# “Automatic detection of the angle between the long axis of the aorta and the valve plane” -> “Automatic detection of the angle between the long axis of the aorta and the valve plain”
- There are duplicate Figure 1 with identical captions.
- This might be a technical glitch. There is no duplicate figure 1 in our submitted manuscript
- Overall, the captions for the figures require improvement.
- We agree that figure captions can be more descriptive
# Figure one caption changed to “Top- Automatic segmentation of the aortic valve annulus with three hinge points, these points are most on the left ventricular outflow tract side for each cusp (left) and left ventricular outflow tract plain, 2-4 mm below and parallel to the aortic valve plain (right). Bottom- Automatic detection of the angle between the long axis of the aorta and the valve plain.”
- The introduction lacks clarity regarding the contributions of this review paper. Utilizing bullet points in the introduction might aid in highlighting the paper's contributions.
- We agree that the goal of the article could have been better spelled out in the introduction, using bullet points. The last paragraph of the introduction reads as follows “In this article, we aim to highlight the growing role of artificial intelligence techniques and algorithms in TAVR procedures, specifically in the following areas:
- Making a diagnosis of severe aortic stenosis
- Patient selection
- Procedural planning and execution
- Predicting post-procedural morbidity and mortality
We also added a comprehensive limitations section, cautioning against the current optimistic hype for the future role of artificial intelligence in clinical medicine, as it may lead to the premature adoption of unvalidated artificial intelligence algorithms in real-world settings, potentially compromising patient safety and quality of care. We also highlight the current challenges and future directions for the role of artificial intelligence in TAVR procedures.”
- In "A major issue in the echocardiography lab is the interobserver variability which arises due to differences in experience, training, and subjective judgment between readers" I would advise clinicians and field experts instead of general readers.
- We think that this statement is common knowledge to clinicians, which constitute the bulk of the journal readers. We agree that this would not make much sense to a public audience.
- I'd recommend enhancing the "Future Directions" section.
- We agree that this section needed enhancement. It now reads as follows: “In the future, the models described in this manuscript will be validated and refined at more TAVR centers. Rigorous validation, regulatory scrutiny, and close collaboration between artificial intelligence developers and healthcare professionals is needed to ensure that artificial intelligence technologies are effectively integrated into clinical practice. Eventually, deep learning models that integrate clinical data with imaging ones -from echocardiography and computed tomography- and conclude the feasibility of the procedure, appropriate valve size, and associated risk for different complications, might be developed.”
- This manuscript lacks a conclusion section, a crucial part that allows authors to emphasize the contributions and motivations driving the necessity of the review paper.
- We have added the following conclusion: “In conclusion, the advanced artificial intelligence algorithms described in this paper are promising tools that have and would further enhance the planning, execution and postoperative follow up of TAVR procedures. The authors also believe that the nuanced clinical judgment by skilled physician teams will remain irreplaceable.”
Reviewer 3 Report (Previous Reviewer 4)
Comments and Suggestions for Authors
i am satisfied that you have addressed the suggested areas of improvement offered by the reviewers.
thank you.
Author Response
No issues to address. Thank you for reviewing.
Round 2
Reviewer 2 Report (Previous Reviewer 3)
Comments and Suggestions for Authors
The authors deserve commendation for their dedicated efforts throughout the manuscript, and their response is notably comprehensive. However, the manuscript still exhibits numerous gaps.
The introduction falls short of establishing a clear context, lacking essential background information. The narrative is unclear, and although the acknowledgment of Machine Learning (ML) and Deep Learning (DL) as established fields is present, their integration into the broader context remains unclear. The manuscript lacks a cohesive storyline and fails to provide a comprehensive overview encompassing the topic, identified gaps, proposed solutions, and future directions drawn from the literature. Moreover, it lacks well-clustered and robust information that could serve as a valuable resource for other researchers seeking knowledge in this area.
Author Response
- The introduction falls short of establishing a clear context, lacking essential background information. This comment is nonspecific. We believe the background has sufficient information, flows nicely and clearly outlines what the manuscript is about.
- The narrative is unclear, and although the acknowledgment of Machine Learning (ML) and Deep Learning (DL) as established fields is present, their integration into the broader context remains unclear. The manuscript lacks a cohesive storyline and fails to provide a comprehensive overview encompassing the topic, identified gaps, proposed solutions, and future directions drawn from the literature. In the current version of the manuscript, there are sections covering the gaps and future directions of this field, including specific ideas for improving the process.
- Moreover, it lacks well-clustered and robust information that could serve as a valuable resource for other researchers seeking knowledge in this area. Interestingly, the other reviewers' assessment is the other extreme of this one. Both other reviewers found the article very helpful with a plethora of valuable information that -by intention- are not too technical or elaborate. The article is rather a comprehensive overview of the current state of the art and the future direction of the field with a necessary caution against the current hype surrounding artificial intelligence.
This manuscript is a resubmission of an earlier submission. The following is a list of the peer review reports and author responses from that submission.
Round 1
Reviewer 1 Report
Comments and Suggestions for Authors
OVerall I really liked this paper, it's an edge topic that will be the next big thing, therefore the readers should be aware of that.
The paper is sound and correct in every aspect, I would however suggest to rearrange it to be have a more reading friendly format, with a nice visual abstract and a clearer subdiviosn in chapters. It is probably better to shorten it by one fourth or very few readers will reach the end.
The methodology is clear and the conclusion are supported by the text of the paper, overall is a good job.
Reviewer 2 Report
Comments and Suggestions for Authors
The article explores the evolving role of AI in TAVR procedures and its potential to enhance patient selection, procedural planning, post-implantation monitoring, and overall patient outcomes. Furthermore, it addresses the current challenges and outlines future directions in the implementation of AI within this context.
As a review paper, this article investigates a substantial amount of literature, but the presentation appears to be overly intricate. It is recommended to present and compare numerical data from the surveyed literature in a more concise manner using tables.
It is also suggested to organize the studies investigated in this paper into comparative tables, covering various aspects such as research topics, sample sizes, technological achievements, etc. The current presentation makes it challenging for readers to grasp the trends in the development of this field.
Reviewer 3 Report
Comments and Suggestions for Authors
The manuscript has an ambitious goal to provide an overview on Artificial intelligence in Transcatheter Aortic Valve Replacement. While I applauded the authors for the work done in the manuscript, I do not believe that the paper is publishable in its current form. There are many reason and many of them are listed below:
The literature review has many gaps. The paper has not clear frame with a huge gap in addressing the AI in Transcatheter Aortic Valve Replacement. The mythology of this paper has to be improved following an academic rigor on writing a review paper. This paper lac in reference and where the authors state the AI capability they do not report reference. I suggest a better and rigorous revision of the state of art in this filed, and adding tables doing a comparing may help. Where the authors report AI models, more info about the structure is important. I do not see a clear improvement of the current state of art on having published this manuscript as it is. However, with more effort, following guideline for review paper editing and an in depth revision of the state of current literature can be useful to improve the overall content and make the paper useful for the comunity.
Comments on the Quality of English LanguageModerate corrections.
Reviewer 4 Report
Comments and Suggestions for Authors
i am grateful for the opportunity to review your manuscript.
i note that your manuscript is submitted to the journal as a Review and not as a piece reflecting original empirical research.
i note also the threefold purpose of your review, as elaborated in the third paragraph of your introduction.
given these caveats, i am happy to endorse publication of your manuscript, as a Review, in this journal.
your manuscript is timely and clearly structured, and i appreciate the care you take to define key terms in the literature, so as to establish common ground between yourselves as authors and your potential readership.
